# MatchVIT: Light-weight Vision Transformer with Matching Separable Self-attention

## Abstract

Vision Transformers (ViTs) have emerged as a powerful architecture in various vision tasks. ViTs process images as sequences of patches and capture long-range dependencies through Multi-Head Self-Attention (MHSA). Hybrid CNN-ViT architectures further enhance performance by integrating the local inductive bias of CNNs with the global contextual information of ViTs. However, the quadratic complexity of self-attention limits its efficiency as the number of tokens rises. Separable Self-Attention (SSA) in MobileViTv2 reduces computational overhead by aggregating contextual information into a single vector and applying the vector to all tokens. Despite this improvement, SSA exhibits limitations compared to MHSA, including extracting only a single level of features, and lacking the ability for tokens to selectively acquire relevant information. These shortcomings further confine the performance of SSA.

To address these issues, we propose MatchViT as a novel hybrid CNN-ViT model. In MatchViT, we introduce Matching Separable Self-Attention (MaSSA), which employs multi-head processing and matching mechanism to enable tokens to individually gather information across hidden tokens. Moreover, Context-gated FFNs in MatchViT leverage the information gathered in MaSSA for enhanced performance. By adopting MaSSA and context-gated FFN, MatchViT achieves a 1%–3% accuracy improvement in Image Classification tasks compared to various other vision models with identical MACs. Other experimental results demonstrate that MatchViT overcomes shortcomings in MobileViTv2, achieving superior accuracy with low computational costs across diverse vision tasks.

## 1 Introduction

Vision Transformers (ViTs) Dosovitskiy et al. (2021), as a powerful alternative to Convolutional Neural Networks (CNNs), have received extensive attention for their outstanding performance across various vision tasks. Based on transformer architecture Vaswani et al. (2017), ViT partitions images into non-overlapping patches and learns features in the form of patch sequences. The core components of ViTs include Multi-Head Self-Attention (MHSA) and Feed-Forward Networks (FFNs). However, the complexity of self-attention exhibits a quadratic relationship with the number of tokens. Therefore, transformers requires extensive computational resource.

To address the computational inefficiency of self-attention, numerous approaches have been proposed. Some researchers have adopted sparse attention to enhance efficiency, including local attention, window attention, and element-wise sparse attention, and the hybrid method of attention mentioned above Liu et al. (2021); Qin et al. (2022a); Deochake et al. (2022). Another solution is linear attention, which decomposes activation functions and replaces the kernel function in it Katharopoulos et al. (2020); Choromanski et al. (2021); Qin et al. (2022b); Han et al. (2024); Qin et al. (2022a). Linear attention calculates the multiplication between keys and values in advance, thus avoiding producing a large attention matrix. Beyond sparse attention and linear attention, there are still various methods to scale down the complexity. MobileViTv2 Mehta & Rastegari (2023) introduced Separable Self-Attention (SSA), aggregating contextual information in a single vector and applying it to all output tokens. MobileViTv2 achieves a 1% higher accuracy than MobileViT while operating significantly faster on mobile devices.

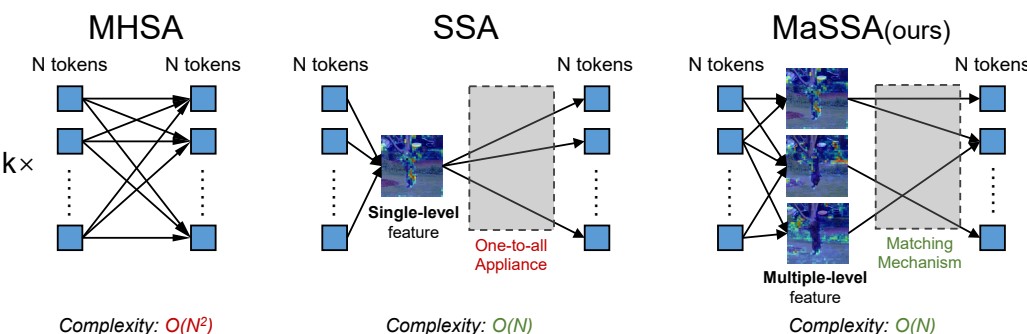

Figure 1: The illustration of various attention methods. SSA processes solely single-level features and delivers the same information to all tokens, thereby limiting the performance of models.

Despite the great trade-off between efficiency and accuracy achieved by MobileViTv2, the SSA mechanism still exhibits inherent shortcomings, presented in Fig. 1. Multi-head Self-attention allows Transformers to process information of multiple levels in parallel with $k$ heads, enhancing the ability to extract features in the context. In contrast, feature with only a single level is extracted and processed in SSA. Furthermore, the attention maps of MHSA reveal that each token requires distinct information. However, all tokens in SSA receive the same information, which further limits the performance of models.

Based on these observations, we propose MatchViT to tackle these drawbacks. MatchViT is a hybrid CNN-ViT model, and mainly consists of Matching Seperable Self-Attention(MaSSA) and context-gated FFN. Compared with SSA, MaSSA utilizes multiple hidden tokens to process information in parallel. Moreover, by conducting the matching process between input tokens and hidden tokens, MaSSA allows each token to obtain information selectively. Context-gated apply features extracted in MaSSA as the gate unit in FFN to further improve the performance. Experiments on MatchViT with different vision tasks show that MatchViTs have overcome the shortcomings and boost the performance with low computational costs.

The contribution of this paper is summarized below.

- We propose Matching Separable Self-Attention (MaSSA) as an effective linear-attention method to learn multiple levels of features in the context and apply to correspoding tokens. The match process allows models to learn both local inductive bias and long-range dependencies.

- We design an FFN components to reuse the information learned in MaSSA. The extra information in MaSSA helps context-gated FFN to extract token-specific information. Context-gated FFN improve the accuracy of the model with almost the same parameter counts and MACs.

- We introduce MatchViTs based on MaSSA and context-gated FFN, and conduct experiments on various vision tasks. The experimental results demonstrate that MatchViTs obtain high performance on various vision tasks while keeping parameter counts and MACs at a low scale.

The rest of the paper is organized as follows. Sec. 2 sorts out the related works, including CNN, ViT, and linear attention. Sec. 3 demonstrates the detailed design of the MatchViT. Experimental results are presented in Sec. 4 and conclusions are given in last section.

## 2 RELATED WORK

**Light-weight CNNs.** CNN architectures is wildly used in many vision tasks, including classification, segmentation and detection. ResNet He et al. (2016) is one of the most common-used CNN architectures for vision tasks. The design of residual connections helps ResNet to expand into deeper layers. Other CNNs increase the parameter counts and redesign the model to achieve better performance, including DenseNet Huang et al. (2017) and SqueezeNet Iandola et al. (2016). However,

the large amount of parameters and FLOPS counts of these CNNs restrict the application scenario in many cases. Some researchers are looking for light-weight CNN architectures, which have less resource usage but perform well in vision tasks. MobileNet family is a series of light-weight CNN models. MobileNetv1 Howard et al. (2017) and MobileNetv2 Sandler et al. (2018) reduce the parameters and FLOPS counts by decomposing the convolution to stack of pointwise and depthwise convolutions. MobileNetv3 Howard et al. (2019) aims to search for a better architecture and propose SE Block. ConvNext Liu et al. (2022) replaces the multi-head self-attention in Transformer to a $7 \times 7$ convolution and achieves better performance than traditional CNN models.

**ViT.** Transformer Vaswani et al. (2017) is originally designed for natural language processing tasks. Dosoviskiy et al. Dosovitskiy et al. (2021) divide images to several non-overlapping patches and apply pure Transformer on the patch sequence. The model is named Vision Transformer (ViT). The results show that ViT can achieve CNN performance under the training of large image datasets. DeiT Touvron et al. (2021) extends ViT using distillation approach, and provides a set of parameters for faster and better ViT training process. Inspired by the pyramid structure of CNNs, PVT Wang et al. (2021; 2022) progressively shrinks the output resolution to reduce the resource consumption. Ze Liu et al. Liu et al. (2021) propose shifted window based self-attention in Swin Transformer to simplify the computational complexity.

**Linear Attention.** Traditional self-attention in Transformer requires high complexity to compute attention matrix. Linear Attention Katharopoulos et al. (2020) lowers the complexity by replacing the kernel function and decomposing matrix multiplication. Performer Choromanski et al. (2021) redesigned the kernel function based on random orthogonal feature. Cosformer Qin et al. (2022b) adopts kernel decomposition in Linear Attention and introduces additional cos-based terms to encode the positional information. Agent Attention Han et al. (2024) introduces agent token to divide the direct interaction between queries and keys. TransNormer Qin et al. (2022a) combines the advantages of local and linear attention, adding normalization to limit the unbound of gradient in linear attention.

In general, hybrid architecture of CNN and ViT obtains great performance in computer vision tasks, while the model size and computational costs remains low. The linear transformer further reduces the complexity of MHSA and accelerate the training and inference process.

## 3 METHODOLOGY

We propose MatchViT, a hybrid architecture of CNN and ViT inspired by the MobileViTv2 architecture. MatchViT is a hybrid architecture of CNN and ViT, which combines the advantages of inductive bias in CNNs and ViT's long-distance information dependency. MatchViT adopts Match separated self-attention to reduce the computational complexity of MHSA from $O(N^2)$ to $O(N)$, where $N$ refers to number of tokens.

In the rest of this chapter, we first briefly introduce Separable Self-attention proposed by MobileViTv2. Then we present the two essential parts of MatchViTs: MaSSA architecture and context-gated FFN.

### 3.1 MOTIVATION OF MATCHVIT

The Motivation of MatchViT is based on an existing method of linear attention, namely Separable Self-Attention (SSA). SSA is a low-computation attention mechanism proposed in MobileViTv2 Mehta & Rastegari (2023). It reduces the computational complexity from square to linear. Specifically, SSA linearly projects the inputs $X \in R^{N,C}$ to three branches: $I \in R^{N,1}, K \in R^{N,C}, V \in R^{N,C}$, where $C$ is the number of channels. A softmax operation is applied to $I$ to produce *Context Score* ($CS \in R^{N,1}$). Next, SSA calculates the distance between *Context Score* and all tokens in $K$ in the vector space. *Context Vector* ($CV \in R^{1,C}$) is then computed as a weighted sum of distance:

$$CV_i = \sum_{N}^{j=1} CS_i * K_{j,i}, \tag{1}$$

where the *Context Vector* represents the global features of the inputs. *Context Vector* shares its information with all tokens in $V$ by element-wise multiplication. Finally, the output result $Z$ is

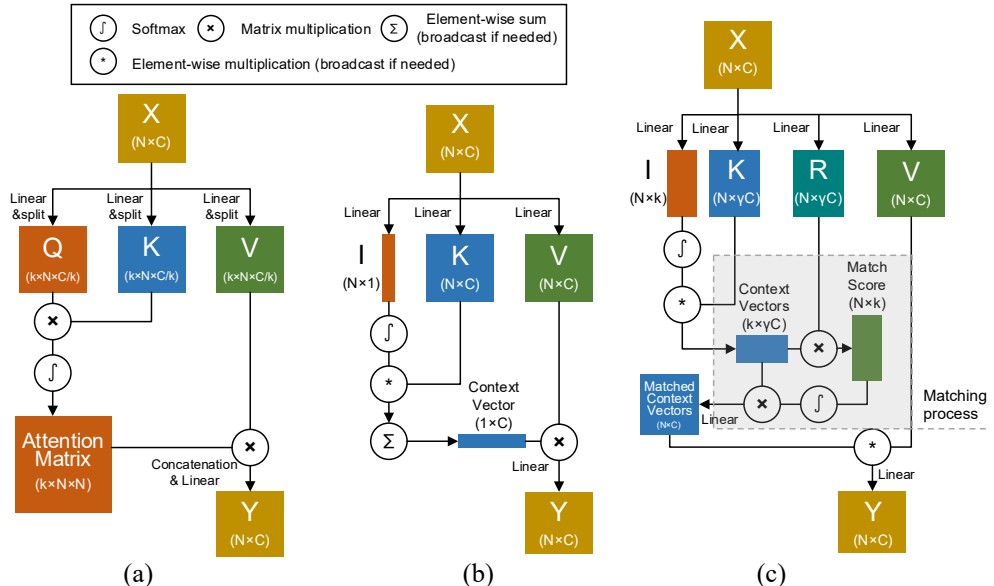

Figure 2: The visualization of MHSA, SSA and MaSSA. (a) MHSA in Transformer. (b) SSA in MobileViTv2. (c) MaSSA in MatchViT. MaSSA extracts feature of multiple level in context vectors, and matching process enable each token to obtain the correspondent features. Transpose operator is omitted for the simplicity of presentation.

obtained through linear projection with weights $W_V$:

$$Y_{i,j} = CV_j * ReLU(V)_{i,j}, \tag{2}$$

$$Z = YW_Y, \tag{3}$$

In general, SSA extracts the features in hidden token and shares the features with all output tokens. The design of hidden token separates the attention and avoids computing a large attention map. However, SSA processes only single-level features and provides uniform information to all tokens, which further restricts model performance. Inspired by the Separable Self-Attention design, we provide our two principal motivations for MatchViT:

**Motivation 1: multiple hidden tokens.** In SSA, the information of every token is reduced to the one hidden token, and generate one context vector. This process leads to only one level of feature extracted, and further limits the model's ability. In contrast, MHSA employs multiple independent attention heads to model different subspaces of information. Inspired by MHSA, we add multiple hidden tokens to self-attention to extract different levels of information in a single layer.

**Motivation 2: mechanism for token matching.** After multiple hidden tokens are added in SSA, the information required from each input tokens are different. A token representing a background pixel in an image may need more low-level texture information than high-level information, and vise versa. Therefore, the mechanism are suppose to determine the matching between the hidden token and input tokens when interacting with the $V$ matrix. We aim to design a matching mechanism, applying different levels of information to the corresponding tokens.

## 3.2 MATCHING SEPARABLE SELF-ATTENTION

Based on the motivations above, we introduce Matching Separable Self-Attention (MaSSA). The visual presentation of MHSA, SSA and MaSSSA is shown in Fig. 2. Based on the first motivation, we expand the shape of matrix $I$ from $R^{N,1}$ to $R^{N,k}$, and the shape of matrix $K$ and $V$ remains to $R^{N,C}$. By multiplication of matrices, MaSSA generates *Context Vectors*, with contains $k$ vectors and multiple levels of features extracted.

Based on the second motivation, we introduce matching mechanism to ensure that each hidden tokens can selectively interact with the output tokens. MaSSA projects the inputs to four branches:

$I \in R^{N,k}, K \in R^{N,\gamma C}, R \in R^{N,\gamma C}, V \in R^{N,C}$. $R$ matrix is derived from the linear projection of the input matrix, and represents the features required for each token. $\gamma$ is a scaling factor to reduce the dimension of *Context Vectors*. Setting $\gamma$ properly reduces the MACs and Parameter counts of model while maintaining the task performance. The matching process of MaSSA is presented in Fig. 2(c).

Specifically, after calculating the *Context Vectors*, the dot-product between *Context Vectors* is performed with the matrix $R$ to obtain the distance between each hidden and each output token. It is followed by a softmax operation to produce *Matching Score*. *Matching Score* is then multiplied with *Context Vector* to obtain *Matched Context Vector*. Finally, element-wise multiplication between *Matched Context Vector* and matrix $V$ is calculated and is followed by a linear projection to obtain the output result. MaSSA enlarges extracted features to multiple levels, and enables features to apply on corresponding tokens through the matching process. MaSSA reduces the complexity from $O(N^2)$ to $O(N)$ when $k \ll N$.

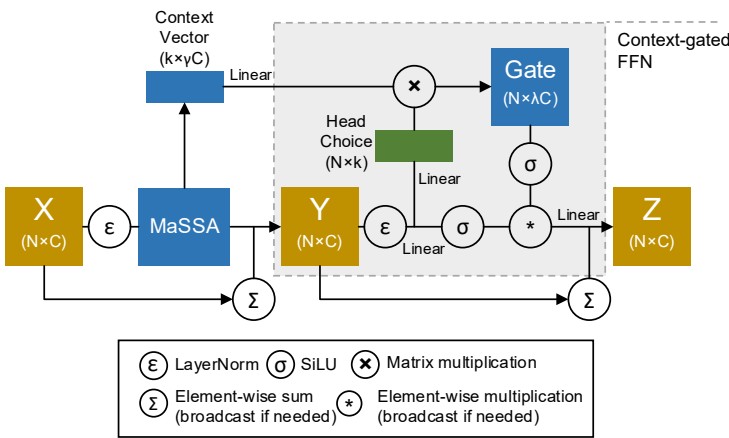

Figure 3: The visualization of context-gated FFN. Transpose operator is omitted for the simplicity of presentation.

### 3.3 CONTEXT-GATED FFN

Feedforward Network (FFN) is another essential part in transformer. FFN layer can be presented as:

$$y = \sigma(xW_1)W_2, \tag{4}$$

where $x$ and $y$ are the input and output of FFN layer, and $\sigma$ is the activation function. $W_1 \in R^{C,\lambda C}, W_1 \in R^{\lambda C,C}$ are weights of linear projection. Bias are omitted for the simplicity. $\lambda$ is the expansion factor of FFN, and is normally set to $2$ or $4$. The LayerNorm and residual connection is omitted for simplification. The FFN projects input features into a higher-dimensional intermediate space and then maps them back to the original dimension. This process allows the model to combine and transform features after self-attention.

Cooperated with MaSSA, we introduce context-gated FFN to reuse the feature extracted before. In context-gated FFN shown in Fig.3, the computation flow is divided into two parts. One path is similar to the original FFN, which consists of linear projection and activation. The other path involves *Context Vectors* in MASSA. Context-gated FFN projects both input matrix and *Context Vectors*, and calculate the dot product The overall procedure of context-gated FFN can be presented as:

$$y = (\sigma(xW_1) * \sigma(xW_k cv W_{cv}))W_2, \tag{5}$$

where $W_k \in R^{C,k}$ is the weight to project input matrix. $cv \in R^{k,\gamma C}$ is the *Context Vectors* in MASSA. $W_{cv} \in R^{\gamma C,\lambda C}, b_{cv}$ are the weight and bias to project *Context Vectors* to intermediate dimension of FFN. Compared with normal FFN, context-gated FFN reuses the *Context Vectors* in MaSSA, which have extracted multiple level of features. Specifically, we detach the *Context Vectors* from its gradients to decouple context-gated FFN from MaSSA while training.

## 3.4 MATCHVIT

Based on our MaSSA and context-gated FFN, we further introduce MatchViT. MatchViT consists of three parts: convolution stem, main structure, and post-processing.

The convolution stem contains one $3 \times 3$ convolution with $1 \times 1$ padding, and an inverted residual block (MV2 block) with the expansion factor of 2. Both parts in the convolution stem are implemented with $2\times$ down-sampling. The main structure of MatchViT contains four stages. Each stage starts with a $2\times$ down-sampling MV2 block, and a MatchViT block is following to learn from the global context. The MatchViT blocks is similar to the transformer block in MobileViTv2, which starts with a separable convolution in Howard et al. (2017) to encode the position information. A stack of transformer encoders are followed after the separable convolution. Each encoder consists of one MaSSA and one context-gated FFN, with residual connection. The expansion factor of MV2 blocks and FFN layers is set to 2. The post-processing part includes a global average pooling and a linear projection to get the classification results.

We provide implementation details of MatchViT in Tab. 1. In the experiment part, we provide three MatchViT models of different size. For MatchViT-XXS, MatchViT-XS, and MatchViT-S, the expansion factor $\alpha$ is set to 0.75, 1, and 1.5.

Table 1: Architecture of MatchViT. *Res.* means the resolution of input feature map in each layer. *In Ch.* represents input channels of each layer, where $\alpha$ is the scale factor for the model. *Head* shows the number of heads in MHSA or hidden tokens in MaSSA. *Rep.* means the repetition of encoder in MatchViT block (MaViT) or MobileViT block (MV2). $\downarrow$ refers to downsample with stride of 2. $\alpha$ is the expansion factor of the entire model.

| Stage | Res. | Type | In Ch. | Head | Rep. |
|---|---|---|---|---|---|
| Stem | 256 | Conv $3 \times 3 \downarrow$ | 3 | - | 1 |
| | 128 | MV2$\downarrow$ | $32\alpha$ | | |
| Stage 1 | 64 | MV2 $\downarrow$ | $32\alpha$ | 4 | 2 |
| | | MaViT | $64\alpha$ | | |
| Stage 2 | 32 | MV2 $\downarrow$ | $64\alpha$ | 8 | 3 |
| | | MaViT | $128\alpha$ | | |
| Stage 3 | 16 | MV2 $\downarrow$ | $128\alpha$ | 16 | 4 |
| | | MaViT | $256\alpha$ | | |
| Stage 4 | 8 | MV2 $\downarrow$ | $256\alpha$ | 32 | 3 |
| | | MaViT | $512\alpha$ | | |
| Post. | 4 | Global Pool | $512\alpha$ | - | 1 |
| | | Classifier | | | |

## 4 EXPERIMENTS

The section of experiments consists of four parts. The first three sections give the detailed results of MatchViT on various vision tasks, including Image Classification, Object Detection, Instance Segmentation, and Semantic Segmentation. The ablation experiments presents the role of each components in MatchViT, and provides visualization of matching process in MaSSA. We implement our models by CVNets Mehta & Rastegari (2023); Mehta et al. (2022), and use its provided scripts for data processing, training, and evaluation.

### 4.1 IMAGE CLASSIFICATION

**Implementation details.** We train MatchViT from scratch for 300 epochs with an effective batch size of 2048. We train and validate the model with ImageNet-1k dataset Deng et al. (2009), containing 1.28 million and 50 thousand training and validation images respectively. We set AdamW Loshchilov & Hutter (2019) as optimizer, with weight decay of $1e^{-2}$. The learning rate is linearly increased from $1e^{-6}$ to $2e^{-3}$ for the first 20k iterations. After that, the learning rate is decayed to $2e^{-4}$ using a cosine annealing policy Loshchilov & Hutter (2017) and label smoothing Szegedy et al. (2016). To reduce stochastic noise during training, we use exponential moving average (EMA)

Table 2: Image Classification results on ImageNet-1k dataset. The results are grouped by the count of MACs. *Params* indicates the parameter counts, and *Top-1* is the top-1 accuracy on ImageNet-1k dataset.

| Model | Type | #Params($\downarrow$) | MACs($\downarrow$) | Top-1($\uparrow$) |
|---|---|---|---|---|
| MobileNetV2-1.0 Sandler et al. (2018) | CNN | 3.5M | 0.3G | 71.8% |
| MobileOne-S0 Vasu et al. (2023) | CNN | 2.1M | 0.3G | 71.4% |
| EMO-1M Zhang et al. (2023) | CNN+SA | 1.3M | 0.3G | 71.5% |
| MobileViTv1-XXS Mehta & Rastegari (2022) | CNN+SA | 1.3M | 0.4G | 69.0% |
| MobileAtt-PVT2B0 Wang et al. (2021); Yao et al. (2024) | LA | 3.5M | 0.6G | 71.5% |
| MatchViT-XXS (ours) | CNN+LA | 3.6M | 0.3G | 74.8% |
| MobileViTv2-0.5 Mehta & Rastegari (2023) | CNN+LA | 1.4M | 0.5G | 70.2% |
| MobileViTv3-0.5 Wadekar & Chaurasia (2022) | CNN+LA | 1.4M | 0.5G | 72.3% |
| PVT-v2-B0 Wang et al. (2022) | SA | 3.7M | 0.6G | 70.5% |
| MobileAtt-PVT2B0 Wang et al. (2021); Yao et al. (2024) | LA | 3.5M | 0.6G | 71.5% |
| MobileViG-T Munir et al. (2023) | CNN+GNN | 5.2M | 0.7G | 75.7% |
| MatchViT-XS (ours) | CNN+LA | 6.2M | 0.6G | 77.6% |
| MobileViTv2-1.0 Mehta & Rastegari (2023) | CNN+LA | 2.9M | 1.0G | 75.6% |
| MobileViTv3-1.0 Wadekar & Chaurasia (2022) | CNN+LA | 3.0M | 1.0G | 76.6% |
| Agent-DeiT-T Touvron et al. (2021); Han et al. (2024) | SA | 6.0M | 1.2G | 74.9% |
| MobileOne-S2 Vasu et al. (2023) | CNN | 7.8M | 1.3G | 77.4% |
| EfficientFormer-L1 Li et al. (2022) | MF | 12.3M | 1.3G | 79.2% |
| SLAB-DeiT-T Touvron et al. (2021); Guo et al. (2024) | LA | 6.2M | 1.3G | 73.6% |
| SLAB-PVT-T Wang et al. (2021); Guo et al. (2024) | LA | 13.4M | 1.9G | 76.0% |
| Agent-PVT-T Wang et al. (2021); Han et al. (2024) | SA | 11.6M | 2.0G | 78.4% |
| Swin-2G Liu et al. (2021) | SA | 12.8M | 2.0G | 79.2% |
| MatchViT-S (ours) | CNN+LA | 13.5M | 1.3G | 80.0% |

T & B (1992). Advanced data augmentation includes Mixup Zhang et al. (2018), CutMix Yun et al. (2019), color jitter, auto-augmentation, and random erasing Zhong et al. (2020).

**Results.** Tab. 2 presents the comparison results between MatchViT and multiple variants of light vision models, including CNN (MobileNetV2 Sandler et al. (2018)), ViT (Swin Liu et al. (2021), PVT Wang et al. (2021; 2022)) and hybrid models (MobileViT Mehta & Rastegari (2023); Wadekar & Chaurasia (2022)). The results are grouped by the count of MACs. MatchViTs outperform various vision models at the same level of computational costs. For example, MatchViT-XXS achieves 3.0% more accuracy than MobileNetV2-1.0 with the similar scale. Compared with MobileAtt-PVT-v2-B0, MatchViT-XXS improves the accuracy by 3.4% with almost the same parameter counts and half the GMACs. MatchViT-XS costs half the GMACs of MobileOne-S2 and less parameter counts, while maintaining a high accuracy. At the accuracy around 80%, MatchViT-S saves 35% of computational cost compared with Swin-2G.

## 4.2 OBJECT DETECTION & INSTANCE SEGMENTATION

**Implementation details.** We use MS-COCO dataset to evaluate the performance of MatchViT series on object detection and instance segmentation task. MS-COCO Lin et al. (2014) is a popular dataset for downstream visual task, which contains 117K training and 5K validation images. We select MaskRCNN He et al. (2017) to implement our detection model. We use the parameters pretrained on ImageNet-1K dataset Deng et al. (2009) to initialize MatchViT as backbone. We train models with 12 epochs, with $2e^{-4}$ initial learning rate and $1e^{-4}$ weight decay. The learning rate will decline by $10\times$ at epoch 8 and 11.

**Results.** The results in Tab. 3 demonstrate the effectiveness of MatchViT on downstream vision tasks. In particular, MatchViT-S achieves $AP^{box}$ of 40.5%, outperforming other models with the close scale, including SLAB-PVT-T and EfficientFormer-L1. MatchViT-XS also excels in precision metrics. MatchViT-XS saves nearly half the parameter counts compared to metrics, including Flatten-PVT-T and PoolFormer-S12. Similar results are found in instance segmentation experiments. MatchViT-S achieves the same $AP^{seg}$ metrics of 38.0 with a lower scale. Compared woth

FastViT-SA12, MatchViT-XS uses 44% less parameters to obtain a simliar performance on IS tasks. These results highlight the effectiveness of MatchViT in OD and IS tasks, with competitive computational efficiency.

### 4.3 SEMANTIC SEGMENTATION

**Implementation details.** We use ADE20K dataset to conduct experiment for MatchViT on segmantic segmentation. ADE20K Zhou et al. (2017) dataset includes about 20K training samples and 2K validation images from 150 categories. We use the parameters pretrained on ImageNet-1K dataset Deng et al. (2009) to initialize our MatchViT models. We select Semantic FPN Kirillov et al. (2019) to conduct experiments, with MatchViT as backbone.

**Results.** In Semantic Segmentation (SS) tasks, evaluated on the ADK20K dataset, MatchViT-S achieves an mIoU of 40.8. MatchViT-S matches the performance of PoolFormer-S24 (40.3 mIoU) while using 34% fewer computational resources. MatchViT-XS also performs admirably with an mIoU of 39.0, surpassing MobileViG-M and EfficientFormer-L1. These results underscore the efficiency and accuracy of MatchViT in SS tasks, making it a strong contender for applications requiring precise semantic understanding with limited computational overhead.

Table 3: The experimental results on Object Detection(OD), Instance Segmentation (IS), and Semantic Segmentation (SS) tasks. OD and IS are conducted on MS-COCO dataset, and SS is conducted on ADK20K dataset. We only measure parameters of backbone, without Mask-RCNN or Semantic FPN. Missing data of baseline is marked as n/a.

| Model | Params($\downarrow$) | OD($\uparrow$) | | | IS($\uparrow$) | | | SS($\uparrow$) |
|---|---|---|---|---|---|---|---|---|
| | | $AP^{box}$ | $AP_{50}^{box}$ | $AP_{75}^{box}$ | $AP^{seg}$ | $AP_{50}^{seg}$ | $AP_{75}^{seg}$ | mIoU |
| FastViT-SA12 | 10.9M | 38.9 | 60.5 | 42.4 | 35.9 | 57.6 | 38.1 | 38.0 |
| ResNet18 | 11.7M | 34.0 | 54.0 | 36.7 | 31.2 | 51.0 | 32.7 | 32.9 |
| PoolFormer-S12 | 11.9M | 37.3 | 59.0 | 40.1 | 34.6 | 55.8 | 36.9 | 37.2 |
| Flatten-PVT-T | 12.2M | 38.2 | 61.6 | 41.9 | 37.0 | 57.6 | 39.0 | 37.2 |
| EfficientFormer-L1 | 12.3M | 37.9 | 59.0 | 40.1 | 34.6 | 55.8 | 36.9 | 38.9 |
| SLAB-PVT-T | 13.4M | 36.5 | 59.0 | 39.2 | 34.4 | 55.7 | 36.5 | n/a |
| MobileViG-M | 14.0M | 41.3 | 62.8 | 45.1 | 38.1 | 60.1 | 40.8 | 38.9 |
| FastViT-SA24 | 20.8M | 42.0 | 63.5 | 45.8 | 38.0 | 60.5 | 40.5 | 41.0 |
| PoolFormer-S24 | 21.4M | 41.4 | 63.9 | 44.7 | 38.1 | 61.0 | 40.4 | 40.3 |
| MatchViT-XXS (ours) | 3.6M | 36.6 | 59.4 | 39.9 | 34.2 | 56.9 | 37.2 | 36.1 |
| MatchViT-XS (ours) | 6.2M | 38.9 | 60.2 | 42.8 | 35.8 | 58.2 | 37.7 | 39.0 |
| MatchViT-S (ours) | 13.5M | 40.5 | 61.4 | 43.0 | 38.0 | 59.6 | 39.9 | 40.8 |

### 4.4 ABLATION STUDY & VISUALIZATION

**Ablation Study.** We conduct ablation experiments on components of MatchViT in Tab.4. #Params indicates the parameter counts, and TOP-1 is the top-1 accuracy on ImageNet-1k dataset. MatchViT(SSA/MHSA) means replacing MaSSA in MatchViT to SSA/MHSA. CG-FFN is the abbreviation of context-gate FFN. We choose MatchViT-S as the baseline of ablation studies.

Table 4: Ablation studies on MatchViT, conducted on attention unit and FFN unit.

| Model | #Params($\downarrow$) | MACs($\downarrow$) | Top-1($\uparrow$) |
|---|---|---|---|
| MatchViT | 13.5M | 1.3G | 80.0% |
| MatchViT (SSA) | 10.9M | 1.2G | 78.2% |
| MatchViT (MHSA) | 11.8M | 1.7G | 80.9% |
| MatchViT (w/o CG-FFN) | 12.1M | 1.3G | 79.2% |

According to the results, the use of the MaSSA helps MatchViTs achieve higher accuracy with slightly higher parameter counts and computational costs, compared with SSA. In contrast, MatchViT with MHSA surpasses the original MatchViT by 0.6% accuracy, whereas at the cost of 1.4$\times$ higher MACs. The quadratic complexity of the MHSA significantly increases the costs of

computing attention at the shallow layer, where the resolution of feature maps is relatively high. MatchViT with MaSSA holds a great balance between accuracy and efficiency. Moreover, context-gated FFN further improves the task performance of MatchViT by $0.8\%$. Reuse of the learned information in *Context Vector* helps context-gated FFN to enhanced ability to extract information.

**Visualization.** To better explain how MaSSA works, we provide visualization of matching process in MaSSA. Fig.4 illustrates the heatmap of *Context Score* and *Match Score* in MaSSA at the resolution of $32 \times 32$ in shallow layers and $8 \times 8$ in deep layers. The input image in Fig. 4 (a) has simple background and objects, and Fig. 4 (b) presents the situation that input image has complex background and objects.

From Fig. 4, we can observe that in shallow layers ($32 \times 32$ resolution), each head is focused on different area of the image in *Context Score* line. In Fig. 4(a), Head 2 is concentrated on the single object in the foreground, and the other heads focus more on areas of background. While in Fig. 4(b), concentrated on various objects in the foreground. The heatmaps of *Match Score* also prove that the information applied to tokens is biased, and is correspondent to the focused areas in *Context Score*. However, in deep layers ($8 \times 8$ resolution), the context score of every head is convergence to the center area. The distribution of *Match Score* in deep layers is more irregular compared to shallow layers. Generally, the visualized results indicate that MaSSA does allow tokens to obtain information of different levels, enhancing the performance of MatchViT.

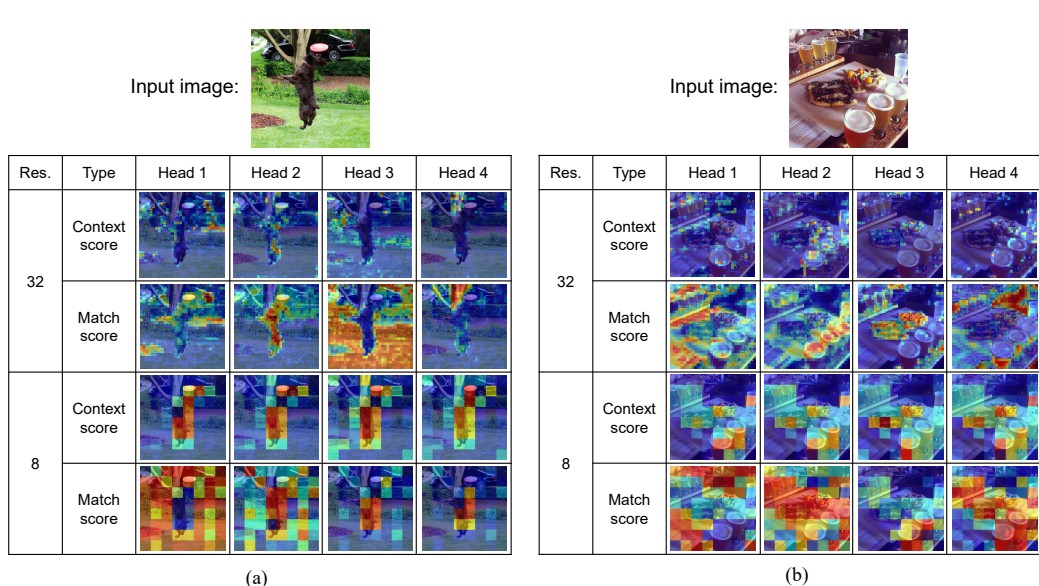

Figure 4: The heatmap of *Context Score* and *Match Score* in MaSSA in different strides. *Res.* refers to resolution of input feature map. (a) Input image has simple background and object. (b) Input image has complex background and object. The visualization results validate that MaSSA is capable of extracting multiple levels of features , and each token selectively obtain different features through the matching mechanism.

## 5 CONCLUSION

In this paper, we propose a series of new light vision models called MatchViTs. MatchViT is a hybrid CNN-ViT model, adopting MaSSA and context-gated FFN to achieve higher accuracy at low computational costs. MaSSA is a novel method based on separable self-attention, learning multiple levels of features by match process. Moreover, we design context-gated FFN to fully exploit the extracted information in MaSSA. By adopting MaSSA and context-gated FFN, our MatchViTs outperform various of vision model on image classification and downstream tasks with low computational resource.

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
