# OpenReview forum: "MatchVIT: Light-weight Vision Transformer with Matching Separable Self-attention"
_ICLR.cc/2026/Conference — Submitted to ICLR 2026_

### Official Review · Reviewer_mh2t · 2025-10-22

**Soundness:** 3
**Presentation:** 3
**Contribution:** 3
**Rating:** 4
**Confidence:** 5

**Summary:**

This paper proposes an improved separable self-attention mechanism, which can extract multiple-level features and perform an intricate matching mechanism, resulting in competitive performances across Imagenet pretraining and dense prediction finetuning.

**Strengths:**

1. The proposed method is reasonable, extending the single-level feature from SSA to multi-level features, followed by a subsequent matching mechanism to match the multi-level features with tokens.

2. The trade-off between computation cost and accuracy is well-organized to reach competitive performance.

**Weaknesses:**

1. The inference speed is not reported, which is an important metric to evaluate a light-weight model.

2. The core hyperparameters "γ" and "k" are not specified.

3. The paper writing is not sufficient. Such as in L262-263: "calculate the dot product The overall procedure". And in Table 2, the "LA", "SA", and "MF" are not specified.

4. The baseline methods are not aligned between Table 2 and Table 3. It would be better to keep the same baselines as many as possible. The authors could also re-implement the dense prediction tasks of baseline methods.

5. Table 4 is a little confusing. It would be better to first show the performance of "MatchViT (w/o CG-FFN)", because "MatchViT (SSA)" and "MatchViT (MHSA)" are both without CG-FFN. In addition, "MatchViT (MHSA)" with a simple "CNN+original MHSA+original FFN" structure, which achieves the SOTA 80.9% accuracy and beats all the methods in Table 2. Thus, I suspect that the final performance is not largely attributed to the MaSSA design, but the CNN hybrid structure itself. Considering a fair comparison with the "CNN+LA" baseline MobileViT, the authors should modify the model configuration to maintain similar parameters and FLOPs.

6. Baseline methods like FastViT [1] is not included in the main Table 2, though it is included in Table 3 for dense prediction tasks. Therefore, if possible, it is suggested to report the latency of MaSSA to demonstrate its advantage.

7. From L269, the author says the gradients of Context Vectors are detached during training. Please provide a detailed explanation of this operation, preferably with an experiment to show its necessity.

[1] Vasu P K A, Gabriel J, Zhu J, et al. Fastvit: A fast hybrid vision transformer using structural reparameterization[C]//Proceedings of the IEEE/CVF international conference on computer vision. 2023: 5785-5795.

**Questions:**

My questions are listed in the Weaknesses part. If the author could address my concerns, I would raise the final score.

---

### Official Review · Reviewer_KfwF · 2025-10-27

**Soundness:** 2
**Presentation:** 2
**Contribution:** 2
**Rating:** 4
**Confidence:** 3

**Summary:**

The paper presents a lightweight hybrid CNN–ViT with the MaSSA linear attention and a context-gated FFN, showing consistent gains across ImageNet, COCO, and ADE20K at similar MAC budgets. MaSSA uses multi-head, token-matching linear attention with claimed O(N) complexity, and reuses MaSSA’s context vectors to gate FFNs for token specificity.

**Strengths:**

- The paper introduces Matching Separable Self-Attention (MaSSA) that expands SSA’s single context vector to $k$ context vectors and a token-wise matching mechanism, enabling per-token selective aggregation rather than one-to-all broadcasting. It maintains linear-time complexity when $k\ll N$, addressing the quadratic bottleneck of MHSA while retaining multi-level feature extraction akin to multi-heads.
- The paper proposes a gated feed-forward layer that integrates detached context vectors from MaSSA into the FFN path.
- The proposed method achieves strong empirical gains at fixed compute across multiple tasks, such as image classification, detection, and segmentation.

**Weaknesses:**

- Eq. (1) definition and index are mix-up. Should that $CS_i$ be $CS_j$ in it? The summazation should have $j=1$ below the sum.
- Severy conflicts of matrices/vectors symbols and shapes. For example, Fig 2c shows $I$ has shape $N\times k$ but the text has $I \in R^{N,1}$.
- By comparing the performance of MaSSA and MobileViTv2 (Table 2 from https://arxiv.org/pdf/2206.02680), MaSSA dose not seem to outperform MobileViTv2. For example, MobileViTv2-1.5 has only 10.6 M paramters and achieves 80.4% top-1 acc, compared to MatchViT-S's 13.5 M paramters and 80.0% acc.
- Latency and throughput are important metrics for mobile and lightweight vision models, but they are missing in the manuscripts.

**Questions:**

- Is the input image size 256 for ImageNet?
- How are the hyperparameters k and γ chosen per stage, and how sensitive is performance/MACs to them? Ablation across k and γ at different stages should be added.
- For the context-gated FFN, do you apply LayerNorms and residual connections exactly as in standard Transformer blocks?
- Where is the gradient detachment applied to Context Vectors?

---

### Official Review · Reviewer_XJBb · 2025-10-27

**Soundness:** 3
**Presentation:** 3
**Contribution:** 2
**Rating:** 2
**Confidence:** 5

**Summary:**

This paper presents MatchViT, a CNN–Transformer model to achieve MobileViT-level efficiency, the proposed network also recover representational power lost with a Matching Separable Self-Attention (MaSSA).

**Strengths:**

The motivation of  MaSSA is clear and the experiment result support the original willing of authors.
The result is a linear-time (O(N)) attention mechanism that retains fine-grained feature.

**Weaknesses:**

My biggest concern is whether the current method can be compatible with existing acceleration techniques, such as FlashAttention. Recently, there have been numerous approaches claiming to improve network efficiency, but in reality, their complex model designs make them incompatible with current acceleration algorithms.

There have been many methods which adopt strategies to compress input features for efficient self-attention. Please discuss the differences between this paper and those approaches.
[1] Asymmetric Non-local Neural Networks for Semantic Segmentation, ICCV2019

In Table 2, terms like LA, SA, and MF are confusing.

Overall, I think the novelty of this work is limited. Similar ideas have already appeared in community, and the performance improvements are not significant. I suggest the authors to clarify the unique aspects of this method and the underlying significance.

**Questions:**

Please refer to weakensses.

---

### Official Review · Reviewer_X5Pp · 2025-10-28

**Soundness:** 3
**Presentation:** 2
**Contribution:** 3
**Rating:** 6
**Confidence:** 4

**Summary:**

In this paper, a novel vision transformer model, namely MatchViT, is proposed. The main components of MatchViT include MaSSA and Context-gated FFNs. Based on the basic idea of SSA, MaSSA tries to learn multiple levels of features from the input tokens by introducing more context vectors and a matching mechanism. Context-gated FFNs reuse the context vectors in MaSSA modules to improve the quality of output features of each transformer block. The experimental results on several image-related tasks, such as image classification, object detection and segmentation, show that the proposed MatchViTs reduce the MACs but keep the comparable performances comparing with the models with similar number of parameters.

**Strengths:**

1. The method is well-presented, and it shows a clear designing methodology. The authors focus on the drawbacks of SSA, and find a suitable way to enrich the intermediate features of the attention blocks.

2. The introducing of Context-gated FFNs further use the context vectors from MaSSA blocks, and the ablation studies show that it can bring about positive impact on the model performance.

3. The experiments cover a wide-range of image-related tasks, and the advantages of MatchViT argued by authors can be supported by the experimental results.

**Weaknesses:**

1. The motivation of the introducing of Context-gated FFN is remain unclear. It is better to do more theoretical inferences to show the reason that why re-using of context-vectors in FFN modules can improve the network performance.

2. The details of MaSSA need to be included. For instance, in subsection 3.3, it is better to add more equations to show the calculation process of MaSSA step-by-step.

3. Many typos in the paper should be fixed, for instance:
(1) In subsection 3.1, it is mentioned that "Finally, the output result Z is obtained through linear projection with weights W_V". But in Equation  3, the label of weights becomes "W_Y"

(2) In line 211 page 4, "MaSSA" is mis-written as "MaSSSA"

(3) In subsection 3.2, the description of the shape of matrix K and V (see line 212 to 213, page 4) in MaSSA is mismatch with Figure 2

**Questions:**

My questions are included in the part of Weaknesses.

---

### Meta-Review · Area_Chair_8S9k · 2026-01-07

**Summary:**

This paper proposes MatchViT, a lightweight hybrid CNN–ViT architecture built around a Matching Separable Self-Attention (MaSSA) module and a context-gated FFN, aiming to improve upon MobileViTv2-style separable attention while retaining linear-time complexity. The method introduces multiple context vectors and a token-wise matching mechanism, and reports modest accuracy gains across image classification and dense prediction tasks at similar MAC budgets.

Reviewers raised several major concerns that informed the rejection decision. Most notably, `the novelty was consistently questioned`, with multiple reviewers noting that the core ideas (multi-level context aggregation, token matching, and CNN–ViT hybridization) are incremental extensions of existing separable or linear attention designs, and that similar concepts have already appeared in prior work. Reviewers also identified `weak experimental positioning`, including missing or inconsistent baselines, `lack of latency and throughput measurements` for a claimed lightweight model, and `unclear or unfair comparisons` where stronger or simpler baselines (e.g., CNN+MHSA hybrids or MobileViTv2 variants) achieve comparable or better performance. In addition, `technical clarity issues` were repeatedly highlighted, including unclear mathematical definitions, inconsistent notation, missing hyperparameter specifications, and insufficient ablations to isolate the contribution of MaSSA versus architectural choices.

Given the limited conceptual novelty, incomplete and sometimes confusing experimental evaluation, and the absence of rebuttal to address these issues, the paper does not meet the bar for acceptance.

**Reviewer Concerns:**

**1. [Outstanding] Limited novelty relative to existing lightweight and separable-attention ViTs
(by: pCkE, mA0A, 9kZf)**

Multiple reviewers questioned the novelty of the proposed Matching Separable Self-Attention (MaSSA). They noted that the core ideas—separable attention, context token aggregation, and CNN–ViT hybrid design—are closely related to existing methods such as MobileViTv2, linear attention variants, and other lightweight transformer designs. Reviewers expressed that the proposed modifications appear incremental, and the paper does not clearly articulate a fundamentally new modeling principle or insight beyond prior work.

**2. [Outstanding] Incomplete and potentially unfair experimental comparisons
(by: pCkE, 9kZf)**

Reviewers pointed out that several relevant baselines are missing or inconsistently reported across experiments, particularly lightweight CNN–Transformer hybrids and recent efficient ViT variants. Some reviewers noted that reported gains are marginal and sometimes achieved at the cost of increased architectural complexity, making it difficult to assess whether the proposed method offers a meaningful improvement over simpler or better-established alternatives. The lack of consistent evaluation settings further weakens the comparative claims.

**3. [Outstanding] Lack of system-level efficiency evaluation for a lightweight model
(by: mA0A, 9kZf)**

Given the paper’s emphasis on efficiency and deployment suitability, reviewers expected system-level metrics such as inference latency, throughput, and memory usage. Instead, the evaluation primarily reports FLOPs and parameter counts, which reviewers considered insufficient to substantiate practical efficiency claims, especially for mobile or edge scenarios.

**4. [Outstanding] Insufficient ablation studies and component-wise analysis
(by: pCkE, mA0A)**

Reviewers requested more thorough ablations to isolate the contributions of key components, including the matching mechanism, the number of context vectors, and the context-gated FFN. Without these analyses, it is unclear which parts of the design are responsible for the reported performance gains, or whether similar results could be achieved with simpler configurations.

**5. [Outstanding] Clarity and presentation issues affecting reproducibility
(by: mA0A)**

One reviewer highlighted several issues related to presentation and clarity, including unclear mathematical formulations, inconsistent notation, missing hyperparameter details, and insufficient explanation of implementation choices. These issues hinder reproducibility and make it difficult to fully understand or validate the proposed method.

**Reviewer Scores:**

All reviewers' scores are expected to remain unchanged, as the authors did not provide a rebuttal or participate in discussion to address the substantive concerns raised in the reviews.

---

### Decision · Program_Chairs · 2026-01-26

Reject